# Unusual multiscale mechanics of biomimetic nanoparticle hydrogels

Yunlong Zhou [1,2,3,4], Pablo F. Damasceno [3,4,5], Bagganahalli S. Somashekar [6], Michael Engel [3,4,7], Falin Tian [1], Jian Zhu [3,4], Rui Huang [6], Kyle Johnson [8], Carl McIntyre [8], Kai Sun [8], Ming Yang [3,4], Peter F. Green [8,9], Ayyalusamy Ramamoorthy [6], Sharon C. Glotzer [3,4,8] & Nicholas A. Kotov [3,4,8,10,11]

Viscoelastic properties are central for gels and other materials. Simultaneously, high storage and loss moduli are difficult to attain due to their contrarian requirements to chemical structure. Biomimetic inorganic nanoparticles offer a promising toolbox for multiscale engineering of gel mechanics, but a conceptual framework for their molecular, nanoscale, mesoscale, and microscale engineering as viscoelastic materials is absent. Here we show nanoparticle gels with simultaneously high storage and loss moduli from CdTe nanoparticles. Viscoelastic figure of merit reaches 1.83 MPa exceeding that of comparable gels by 100–1000 times for glutathione-stabilized nanoparticles. The gels made from the smallest nanoparticles display the highest stiffness, which was attributed to the drastic change of GSH configurations when nanoparticles decrease in size. A computational model accounting for the difference in nanoparticle interactions for variable GSH configurations describes the unusual trends of nanoparticle gel viscoelasticity. These observations are generalizable to other NP gels interconnected by supramolecular interactions and lead to materials with high-load bearing abilities and energy dissipation needed for multiple technologies.

[1] School of Biomedical Engineering, School of Ophthalmology and Optometry and Eye Hospital, Wenzhou Medical University, Wenzhou, Zhejiang 325011, China. [2] Wenzhou Institute of Biomaterials and Engineering, Chinese Academy of Sciences, Wenzhou 325000, China. [3] Department of Chemical Engineering, University of Michigan, Ann Arbor, MI 48109, USA. [4] Biointerfaces Institute, University of Michigan, Ann Arbor, MI 48109, USA. [5] Department of Cellular and Molecular Pharmacology, University of California, San Francisco, CA 94143, USA. [6] Biophysics and Department of Chemistry, University of Michigan, Ann Arbor, MI 48109, USA. [7] Institute for Multiscale Simulation, Friedrich-Alexander-University Erlangen-Nurnberg, Erlangen 91052, Germany. [8] Department of Materials Science and Engineering, University of Michigan, Ann Arbor, MI 48109, USA. [9] National Renewable Energy Laboratory, Golden, CO 80401, USA. [10] Department of Biomedical Engineering, University of Michigan, Ann Arbor, MI 48109, USA. [11] Michigan Center for Integrative Research in Critical Care (MCIRC), Ann Arbor, MI 48109, USA. Correspondence and requests for materials should be addressed to Y.Z. (email: zhouyl@wibe.ac.cn) or to N.A.K. (email: kotov@umich.edu)

Molecular or nanoscale components are known to form macroscale solids through the process of gelation[1], driven by the formation of covalent and non-covalent bonds between components[2–5]. Gels formed by polymers, proteins, peptides, lipids, surfactants, and micron-sized colloids have been studied and used for decades, but the gel-forming abilities of water-soluble inorganic nanoparticles (NPs) were discovered only recently[6–9]. The many different forces involved in NP–NP interactions tunable by NP size, material, and surface ligands open modern materials engineering methods to expand the spectrum of the mechanical properties of these gels[10,11]. In particular, the organic/inorganic duality of many NPs leads to strong van der Waals attractions between the cores. Simultaneously, supramolecular interactions at the NP interfaces make them protein-mimetic[12] and are expected to offer an additional means for attaining viscoelastic behavior, by contributing both particle reconfigurability and enhanced attractive forces. The resulting hybrid hard core/soft shell behavior is anticipated to deviate from theoretical predictions of gels based on hard spheres and remains largely unknown[13–17]. The conceptual framework for predicting the behavior of NP gels as viscoelastic materials is absent and may not be intuitively derived from other gels. One of the reasons to make experimental and theoretical steps in this direction would be to find NP composites with simultaneously high-viscoelastic storage and loss moduli[18,19]. As for many soft biological tissues made from collagen, laminin, actin, etc., such structural materials are essential in multiple technological areas but difficult to engineer, for instance, prevention of structural aging needed for multiple technologies ranging from electronics to aviation.

The 'blue-prints' for high stiffness-high damping solids typically involves intricate, three-dimensional architectures rather than gels[18,19]. The energy dissipation channels in NP gels will be markedly different from gels made from organic molecules, polymers, proteins, or purely inorganic NPs, such as silica[13,20–22], and offer multiscale engineering pathways and an expended toolbox in this field. For example, the ability of NPs to self-organize in solid state[23,24] engages multiple interactions between NPs, increases energy dissipation and offers property engineering methods specific to NPs, for instance, by external electrical field[24].

In this study, we synthesized NP hydrogels from semiconductor NPs coated with a peptide layer of glutathione (GSH) and report the mechanical behavior of their hydrogels. The viscoelastic properties of gels made from NPs bearing short ligands revealed marked differences compared to other gels, because their mechanical behavior simultaneously engages the interactions of both the NP core and the surface ligands while retaining their reconfigurability. Furthermore, their macroscale mechanical behavior can be rationalized based on a simple computational model accounting for nanoscale structural parameters of the constituent particles and molecular scale interactions of the surface ligands.

## Results

**Preparation of GSH-NP hydrogels**. NP gels were synthesized from CdTe NPs, stabilized by a natural peptide L-glutamyl-L-cysteinyl-glycine (GSH) (Methods section)[25]. The choice of GSH as a NP surface ligand was made based on its ability to form multiple intermolecular hydrogen bonds. We also wanted the stabilizer to be relatively short so that the powerful dispersion forces, or van der Waals (vdW) interactions, between the inorganic cores of the NPs would contribute substantially to the gel formation. These NP gels can also be viewed as biomimetic analogs of dense biological gels found, for instance, in cytoskeleton, where supramolecular interactions at the protein–protein interface are augmented by forces specific to inorganic

materials[12]. Such NP solids combining the strong core-to-core and shell-to-shell interactions were expected to show unusual viscoelastic properties that are reminiscent of many biological gels.

NPs with inorganic core diameters of 2.7, 3.2, 3.7, and 4.0 nm were made by varying the duration of the reflux stage (Supplementary Table 1 and Supplementary Figures 1–3). The four sizes of GSH-CdTe NPs differ from each other by one additional layer of the CdTe atomic crystal structure. Aqueous dispersions of NPs with diameters of 2.7–3.7 nm remained stable after 24 months but those with diameters of 4.0 nm were stable only for several days. While the 4.0 nm NPs were sufficient for some benchmark tests, most of the experimental work was carried out with 2.7, 3.2, and 3.7 nm NPs.

When 2.5 volumes of isopropanol were added to one volume of an aqueous dispersion of NPs, phase separation occurred, and produced a viscous fluid phase (Fig. 1, Supplementary Figure 1). The isopropanol-rich phase was separated by centrifugation at 3000 r.p.m./min and dried under vacuum. The dried NP solid was hydrated by adding 22 w/w % DI water, which transformed it into a voluminous, intensely luminescent gel with a high-quantum yield of about 30 % (Methods section). During the preparation of the gel, we noticed that this material was unusually stiff, yet able to flow under shear (Fig. 1).

The aqueous dispersion of NPs displayed photoluminescence (PL) peaks at 523, 566, 600, and 647 nm, whereas the prepared hydrogels had PL emission peaks at 544 nm (green), 590 nm (orange), 618 nm (red), and 657 nm (deep red), respectively (Fig. 1a–c, Supplementary Figure 4). The redshift of PL emission peaks in the gel state, as compared to the PL peaks of the initial aqueous dispersion of CdTe NPs was attributed to energy transfer between the closely spaced NPs. For brevity, we shall refer to the corresponding hydrogels as Hydrogel-544, Hydrogel-590, Hydrogel-618, and Hydrogel-657. For brevity, most of the discussion that follows is centered on the former three gels as being most representative of the GSH-CdTe NP gels; Hydrogel-657 is used as a test case for essential property trends.

**Viscoelastic properties**. Oscillatory stress/strain tests of the freshly prepared NP hydrogels with different NP sizes revealed that the stress increased linearly with the strain amplitude applied to the sample at the initial stage. Surprisingly, the gel made from the smallest NPs, Hydrogel-544, displayed the highest stiffness and shear modulus for all strains (Fig. 1d, e). We also performed static compressive tests on cylindrical specimens. Similar to hydrogels, the toughness of aerogels composed of smaller NPs are higher than those composed of larger NPs (Supplementary Figure 5). This observation contradicted our expectations and multiple previous studies of NP solids with long organic ligands[26–29], as well as gels from various inorganic particles without distinct organic shells[30]. The reasons for such unexpected particle size dependence of viscoelastic properties should be related to the combination of strong attraction interactions between the NPs while retaining their reconfigurability that enables efficient energy dissipation.

At high strains, distinct nonlinear behavior was observed as the gels began to flow (Fig. 1d, e, g, h, j, k). This finding is unusual because the volumetric ratio of the soft organic shell made from GSH versus the hard CdTe core was the largest (3.7:1) for Hydrogel-544, as compared to 2.8:1 for Hydrogel-590, and 1.9:1 for Hydrogel-618 (Supplementary Figure 6). The critical strain whereupon the gel starts to flow can be determined by the flex point in the strain dependences for the storage modulus $G'$ and the loss modulus $G''$ (Fig. 1e, h, k). This flow point is at about 1 % strain for Hydrogel-544, while Hydrogel-590, and Hydrogel-618

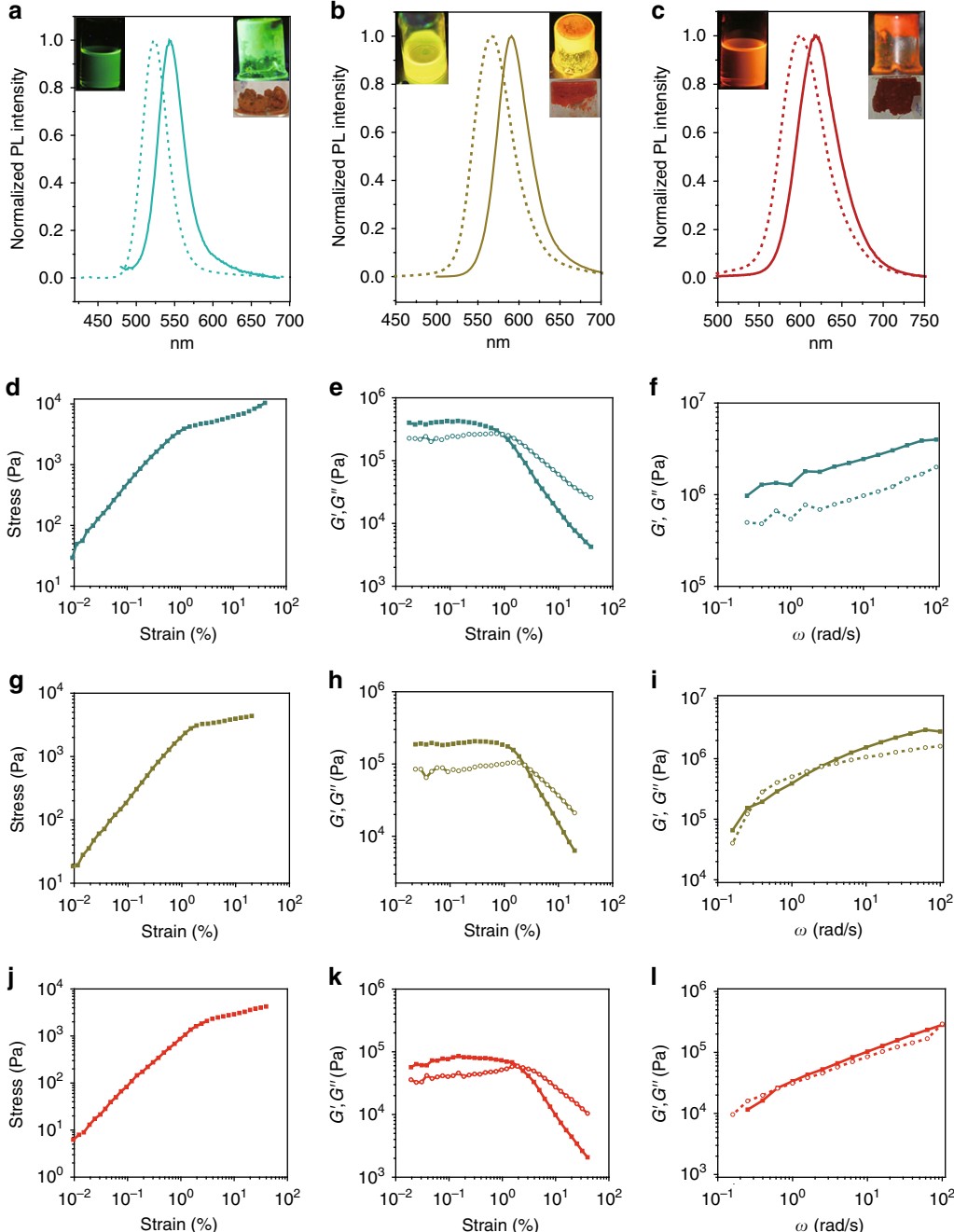

**Fig. 1** Optical and mechanical characterization of hydrogels. Optical images and fluorescence spectra of CdTe NP solutions (dashed line) and hydrogels (solid line), Hydrogel-544 (**a**), Hydrogel-590 (**b**), and Hydrogel-618 (**c**). Measurement of oscillatory stress/strain of Hydrogel-544 (**d**), Hydrogel-590 (**g**), and Hydrogel-618 (**j**). Measurement of continuous step moduli/strain of Hydrogel-544 (**e**), Hydrogel-590 (**h**), and Hydrogel-618 (**k**). Each measurement was performed at least twice on two different disc specimens from the same sample. Rheological dynamic oscillatory frequency sweep tests of Hydrogel-544 (**f**), Hydrogel-590 (**i**), and Hydrogel-618 (**l**). The solid data point and dark color lines in **e**, **f**, **h**, **i**, **k**, and **l** correspond to the storage moduli $G'$ ($\omega$), which is associated with gel elasticity (stiffness). The open data point and light color lines in **e**, **f**, **h**, **i**, **k**, and **l** correspond to the loss moduli $G''$ ($\omega$), which describes energy dissipation in the gel. The shear dynamics for each NP gel was measured from low to high-strain starting at 0.01–20 % at a frequency of 1 Hz or 6.28 rad/s. The rheological dynamic oscillatory frequency sweep measurements were performed with a parallel fixed plate geometry (diameter 25 mm) at a strain value of 0.01 %

begin to flow at about 2 % strain. In addition, the values of the storage moduli drop more rapidly for Hydrogel-544 than for gels from larger NPs. This means that once the structure of the gel is disrupted, the interactions between the smaller NPs are weaker than those of the larger NPs.

Additional knowledge about viscous deformations in these fluids can be obtained from the variation of dynamic mechanical properties with cyclic frequency, $\omega$. The response of Hydrogel-544 (Fig. 1f) is remarkably different from those of Hydrogel-590 and Hydrogel-618 (Fig. 1i, l). Hydrogel-544 reveals a plateau for both moduli and over the entire range of frequencies $G' > G''$. For Hydrogel-590 and Hydrogel-618, both moduli increase with $\omega$ and show a transition from mostly elastic to mostly viscous behavior at $\omega \sim 5$ rad/s when $G' \sim G''$. Hydrogel-657 again shows

lower values of stiffness, shear modulus, storage moduli, and loss moduli compared to Hydrogel-618 (Supplementary Figure 7).

The nearly ideal linear stress–strain response of Hydrogel-544 in Fig. 1d is indicative of the dominance of elastic interactions between NPs at the nanoscale. Such viscoelastic behavior is uncommon for hydrogels because it is typically determined by highly dissipating intermolecular bonds. Moreover, the magnitude of $G'$ for Hydrogel-544 exceeds $10^6$ Pa for some frequencies. These values of storage moduli are considerably higher than those of other hydrogels made from a wide spectrum of other chemical, biological, and nanoscale components with different structural organizations[11,30], in some cases by several orders of magnitude (Supplementary Table 4); such high values of $G'$ support the possibility of simultaneously combining high stiffness and high-energy dissipation in these materials.

The viscoelastic properties of materials can be cumulatively characterized by the viscoelastic figure of merit (VFOM) calculated as $VFOM = |G^*|\tan\delta$, where $G^* = (G'^2 + G''^2)^{0.5}$ and $\tan\delta = G''/G'$. Hydrogel-544 has exceptionally high VFOM with a value of 1.83 MPa at 10 Hz (Supplementary Table 3). Similarly high values of $VFOM = 1.71$ MPa were observed for Hydrogel-590, while for other NP gels made with GSH, CYS, MPA ligands and both CdTe and Au cores ranged from 0.01 to 0.67 MPa in the wide range of frequencies. Even for the smallest values of VFOM in this family of gels, they exceed VFOMs for most advanced NP hydrogels based on niobate nanosheets[2] whose VFOM is < 0.001 MPa or articular cartilage[3] with VFOM ~ 0.001 MPa. Such high VFOMs in NP based gels are attributed to the strong attraction between the NPs while retaining their reconfigurability. Besides practical significance associated with unusually high VFOMs for gels[18], the materials that can combine high values of both $G''$ and $G'$ challenge the fundamental understanding of about the property correlations and how the materials the limits apparent limits specific to particular classes of the materials can be overcome.

**Morphology**. Hydrogel-544 clearly stands out among other hydrogels studied here. Understanding the origin of the unique viscoelastic properties for this and other hydrogels studied here requires better understanding of the gel structure. The freeze-dry process was used to remove constituent water without the disruption of the NP networks. SEM and STEM high-angle annular dark-field (HAADF) images (Fig. 2a–c) revealed that all three hydrogels are structurally similar in that they are comprised of densely packed CdTe NPs (Fig. 2d–f). However, the hydrogels were solid with low-density porous structures indicated by Brunauer–Emmett–Teller (BET) analysis, small-angle X-ray scattering (SAXS), and X-ray tomography (Methods section and Supplementary Figures 8–10). Energy-dispersive X-ray elemental mapping (Supplementary Figure 11) confirmed that the packed NPs are distributed throughout the hydrogels. Also, evaluation of these hydrogels by thermogravimetric analysis revealed that the GSH content was higher than 31.7 w/w % (Supplementary Figure 6), which indicated that most of the stabilizer molecules around the NPs remained in place. Given the similarity of the packing within the hydrogels, the path to understanding the differences between their bulk viscoelastic properties must go through the understanding of the structure and interactions within NP–NP interfaces.

The rheology of NP hydrogels could potentially be understood using previous models of colloidal gels[13,30,31] but these approaches face the well-known problem of combining the description of the NP ligand shell at the molecular level while requiring description of mechanical behavior at the macroscale level. For example, the sticky hard-sphere model is used to describe inert and rigid colloidal systems, in which the only inter-particle forces considered include infinite repulsion if the particles overlap, and a strong attractive interaction on contact[32]. The conceptual make-up of these models make them quite restrictive in terms of the spectrum of mechanical properties they can predict[14]. Most importantly, current colloidal gel models face the problem of describing the gels as having simultaneous and equally significant contributions from both the NP cores and surface ligands that represent multiple scales and complex interactions, especially in the presence of a solvent[12,33].

**Molecular structure of the GSH layer**. The GSH surface ligands are bound to the surface of CdTe NPs via $Cd^{2+}$ ions. We studied this bonding by $^1H$ and $^{113}Cd$ NMR spectroscopy using a 500 MHz spectrometer. First, as a control experiment, the $^1H$ NMR spectra of $Cd_x(GSH)_y$ complexes revealed that the cysteinyl (cys) and the glutamyl (glu) proton resonances of GSH shift to higher frequency as compared to unconjugated GSH (Fig. 3a, b). $^1H$ NMR spectra of all three hydrogels showed two well-separated pairs of proton resonances $C(\alpha)$ (4.47 and 4.61 ppm) and $Q(\alpha)$ (3.68 and 3.58 ppm). We did not observe NMR signals attributable to considerable amount of water in the gels. The existence of pairs indicates the presence of two types of distinct coordination geometries for GSH bonding with $Cd^{2+}$ (Fig. 3c–e). In addition, two-dimensional $^1H$–$^1H$ exchange NMR spectroscopy (EXSY) confirmed that the two bonding types are interchangeable by fast ligand exchange (Fig. 3f, Supplementary Figure 12).

To resolve molecular details of GSH bonding, we analyzed the $^{113}Cd$ chemical shifts, which are sensitive to the $Cd^{2+}$ coordination environment. The $^{113}Cd$ NMR spectrum of Hydrogel-544 shows two distinct cadmium resonances at 324.55 ppm (dominant) and 678.3 ppm (Supplementary Figure 13). For the assignment of these two peaks, we compared the chemical shift values with reported values from the literature[34]. The $^{113}Cd$ peak for the $Cd(S-GS)_4$ complexes is expected at 674 ppm, whereas the $^{113}Cd$ peak for the interchangeable $CdS_2N_3O/CdSN_3O_2$ complexes typically appears at 322 ppm. We attribute the dominant peak at 324.55 ppm to GSH moieties that are simultaneously bound via a covalent bond to Cd atoms and two coordination bonds with $-\underline{N}H_2$ and $-CO\underline{O}$ groups. A similar scorpion-like configuration of GSH has been observed upon its binding to a gold surface[35].

We further refer to this coordination geometry as the three-point bonding (TPB) mode. In TPB model, CdTe might include three or two of Cd atoms coordinating with S, $-\underline{N}H_2$, and $-CO\underline{O}$. To answer this question, we determined the TPB–GSH model by a quantum mechanical semi-empirical calculations using the software package Spartan (Wavefunction Inc., Irvine, CA). (Fig. 3g, Supplementary Figure 14). In contrast, we attribute the signal at 678.3 ppm to the GSH in a standard single-point bonding (SPB) mode that could also be described as the 'crew-cut' configuration (Fig. 3h). Bonding at three points versus one point on the NP surface makes TPB–GSH acquire an atomic configuration that is nearly parallel to the surface (Fig. 3g), as compared with the vertically aligned SPB–GSH (Fig. 3h). Furthermore, we hypothesize that making three bonds with the surface rather than one likely requires the molecule to be anchored at an edge or a corner.

This hypothesis can be evaluated using NMR spectroscopic techniques that afford quantification of the number of GSH molecules in the two bonding modes by measuring the relative intensities of signals corresponding to the $C(\alpha)$ protons, which resonate at 4.47 and 4.61 ppm. Accordingly, the ratios of GSH in the TPB configuration (TPB–GSH) to GSH in the SPB configuration (SPB–GSH) for Hydrogel-544, Hydrogel-590, and

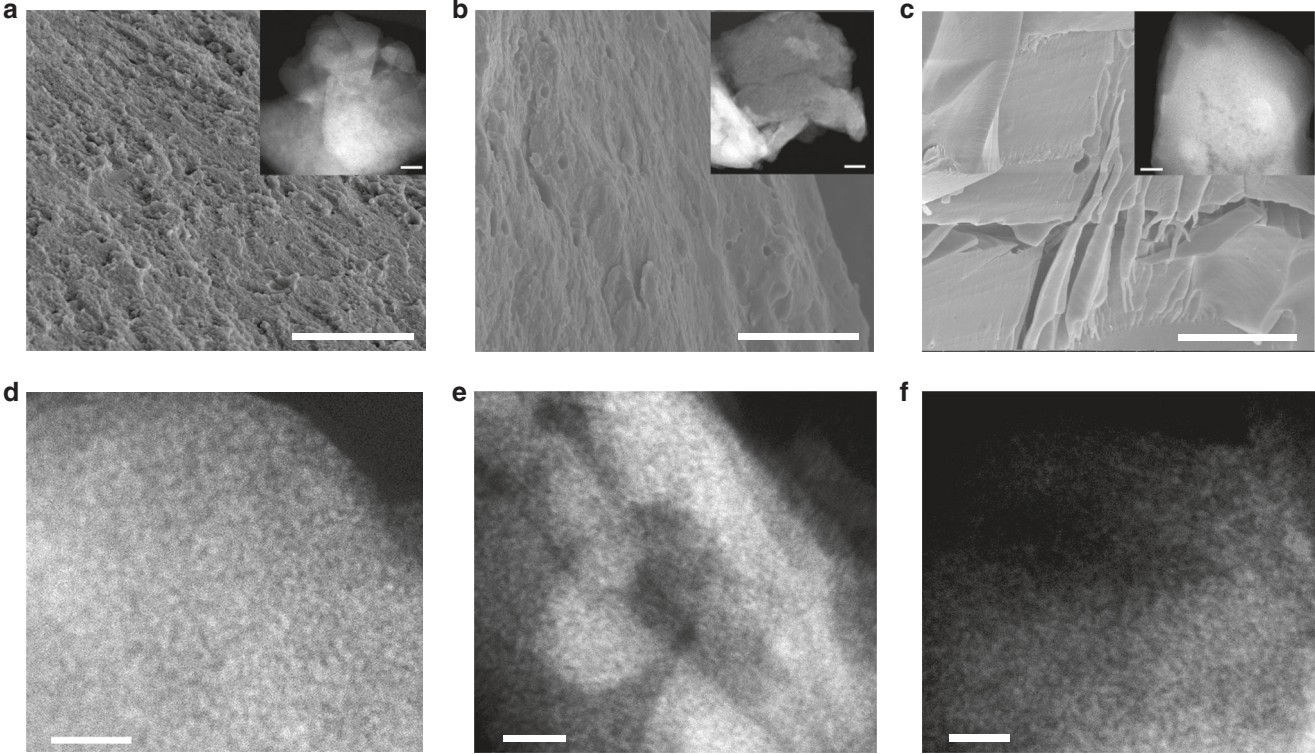

**Fig. 2** Electron microscopy of hydrogels. **a**–**c** SEM and STEM HAADF images of hydrogels. The scale bar of inset HAADF images is 50 nm. **d**–**f** High-magnification HAADF images of hydrogels. The images show Hydrogel-544 (**a**, **d**), Hydrogel-590 (**b**, **e**), and Hydrogel-618 (**c**, **f**). Scale bar in **a**–**c**: 5 μm, Scale bar in **d**–**f**: 20 nm

Hydrogel-618 are 1:0.2, 1:0.4, and 1:0.7, respectively. The relative frequency of TPB configurations is largest for the smallest NPs, which can also be seen via X-ray photoelectron spectroscopy (XPS) (Supplementary Figure 15). The dominance of the N–Cd coordination in Hydrogel-544, as evidenced by a peak at 401 eV for N 1s, cannot be detected in Hydrogel-618.

XPS data further demonstrate the difference in the structure of the interface between the NPs in Hydrogel-544 and Hydrogel-618. Hydrogel-544 and Hydrogel-618 display XPS peaks for the Cd 3d line at 404.5 and 411.3 eV, respectively, indicating a marked difference in the electron density around the Cd atoms on the NP surface. The effect of the stabilizer configuration on the hydrogen bonds can be concomitantly observed from chemical shifts and signal broadening in the $^1$H NMR experiment (Fig. 3a–e). The chemical shifts of GSH corresponding to the binding of $Cd^{2+}$ with the functional groups –COOH, –SH, and –NH$_2$ increase from Hydrogel-544 to Hydrogel-590 and Hydrogel-618. This is indicative of a weakening of the hydrogen bonding due to an increase of the electron density on the hydrogen atom.

**Molecular effects on macroscale mechanics of nanoparticle hydrogels.** One might expect that solids made from NPs with higher volume organic stabilizer shells and relatively small-crystalline cores will be easier to deform than those with a larger proportion of inorganic material. However, this is not the case here. Thermogravimetric analysis reveals that weight ratios between GSH and CdTe for Hydrogel-544, Hydrogel-590, and Hydrogel-618 are 0.90, 0.68, and 0.46, respectively. The gel with the smallest NPs, Hydrogel-544, exhibits not only the highest stiffness and highest loss modulus (Fig. 1f), but also the lowest weight ratio of CdTe.

In order to integrate the molecular scale descriptors into a theoretical models suitable for initial assessment of macroscale mechanical properties, we calculated the surface density of TPB–GSH and SPB–GSH from the weight ratios and NMR measurements under the reasonable assumption that all GSH stabilizers reside in shells around the NPs (Methods section). The reasons behind this approach to modeling of mechanical properties are governed by the desire to construct a relatively simple yet descriptive model generalizable to different NPs. More complex models with explicit description of hydrogen bonds would be difficult to parametrize because of strong dependence of hydrogen bonds on the specific configuration of GSH on the surface of NPs. While MD simulations combined with DFT or ab initio modules were successful for biomimetic NP capsids[36] and NP hydration layer[37], the relatively large size of the NPs in our case, large number of ligands on the surface of the NPs and the large number of NPs required to model gel mechanics make this approach difficult to realize at the moment. The model presented here can be used as a stepping-stone toward the goal of relating intrinsic particle properties and macroscopic measurements in hydrogels.

The absolute number of TPB–GSH per NP remains approximately constant at 19.8, 21.9, and 18.6, whereas the same parameter number grows for SPB–GSH from 3.9 to 8.8 to 13.1 (Methods section). We hypothesize that the relatively higher number of TPB–GSH on the surface of smaller CdTe NPs could account for the higher stiffness observed in experiments. This could be so if TPB–GSH interactions yielded stronger NP–NP interactions than SPB–GSH bindings. In order to test this hypothesis, we created a generalized interaction strength ratio parameter quantifying the degree by which TPB–GSH mediated interactions are stronger than SPB–GSH. We then analytically calculated the impact this ratio would have in the strength of an identical system of interacting NPs. Since we do not know the

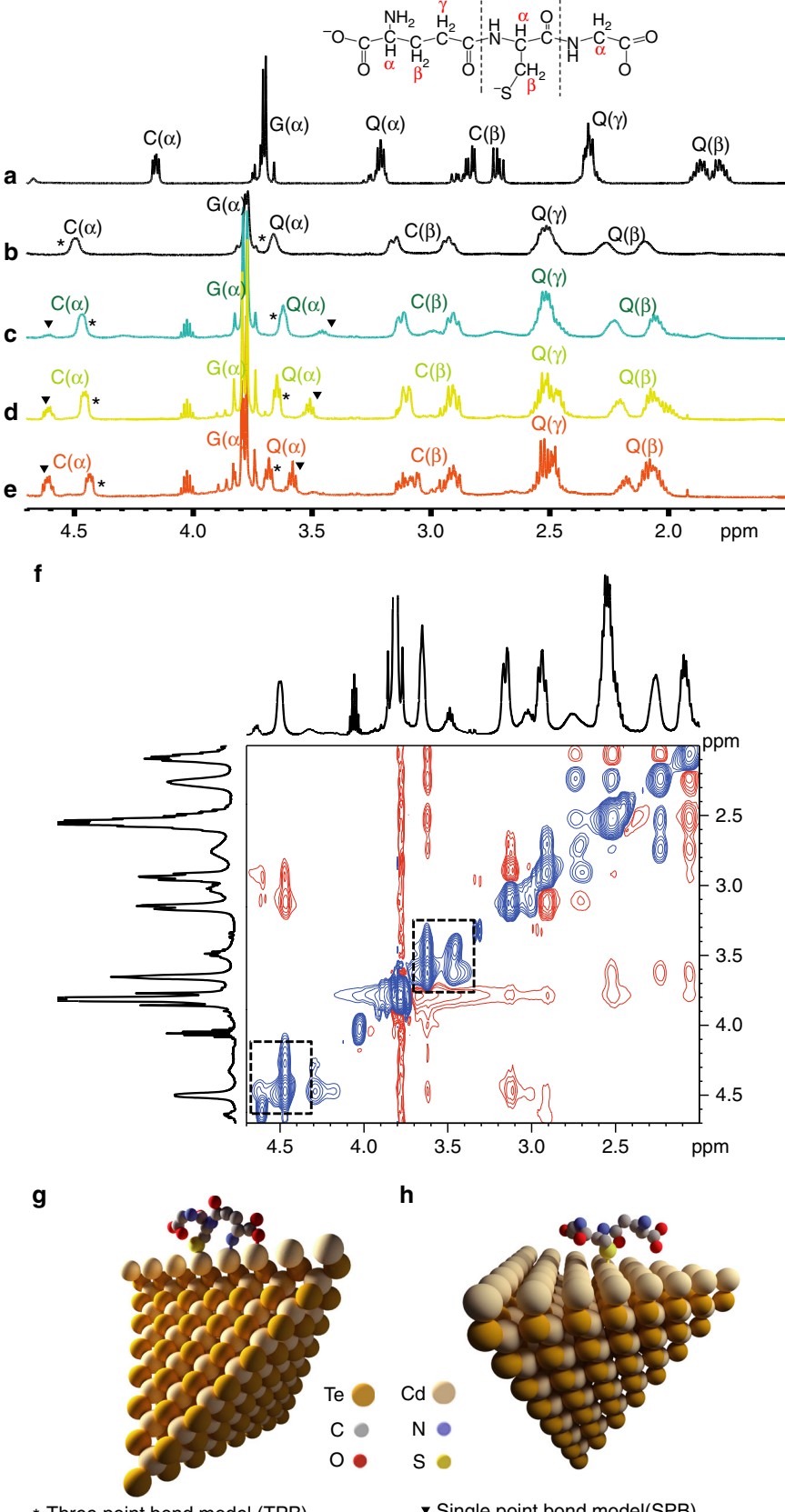

location of these ligands, we measure this impact for both edge-mediated interactions and facet-mediated interactions. Both upper and lower bounds, resulting from these considerations, are shown in Fig. 4a. As it can be seen, if TPB–GSH mediated interactions are twice as strong as SPB–GSH-mediated interactions and primarily located at the NP edges (upper bound), a threefold increase in stiffness of Hydrogel-544 compared to Hydrogel-618 is observed, in agreement with our experimental findings (Fig. 1d). We assumed even the smaller CdTe NPs to have tetrahedral shapes, following evidence from previous works[38,39].

The fact that the number of TBP–GSH molecules is approximately constant despite the changing NP core suggests that they reside preferentially on the edges and vertices, which change less rapidly with changing NP size than surface area. Similarly, the rapid increase in number of SPB–GSH ligands with particle size, suggests a preference for the facets of the inorganic core. These assumptions also make sense in light of recent theoretical findings showing the preference of ligands for vertices, edges, and faces, respectively[38–40], suggesting a mechanism by which the first bound ligands, attached to vertices and edges, have enough room to change configuration from SPB to TPB, whereas latecomers end up stuck in a SPB configuration on the facets (Fig. 4b). A test for this particular set of assumptions is however left for future work. Considering that the nearly horizontal arrangement of TPB–GSH molecules (Fig. 3g) means larger numbers of atoms exposed for hydrogen bonding between NPs, it is reasonable to expect TPB–GSH to result in greater attraction between neighboring NPs.

**Tuning of Gel mechanics using molecular and nanoscale parameters.** A step toward generalization of the observed relationship between the molecular configurations of surface ligands and the macroscale viscoelastic properties of NP hydrogels can be made if similar properties and relationships are observed for gels from similar—but not identical—NPs. More specifically, since: (i) our theoretical model suggests that the different ratio between three-point and single-point bonds are responsible for the drastic changes in gel stiffness, and (ii) the preference of three-point bonds for the NPs corners and edges is a general mechanism for the stabilization of facetted NPs[38–40], we investigate, as positive and negative benchmarks, an additional system of facetted NP stabilized by GSH, and two other systems of CdTe, stabilized this time by two ligands not showing multiple molecular configurations (Supplementary Table 2). The hydrogels made from GSH-stabilized Au NPs (GSH-Au), as well as from cysteine- and mercaptopropionic acid-stabilized CdTe NPs, are denoted as CYS-CdTe and MPA-CdTe hydrogels, respectively.

In general, GSH-Au hydrogels displayed similar mechanical behavior to GSH-CdTe hydrogels and similar molecule structure of GSH on Au NPs with GSH-CdTe (Fig. 5a-c, Supplementary Figure 16). As expected, higher stiffness and energy dissipation was observed for small GSH-Au NPs over large ones (Fig. 5b, c). Moreover, the values of storage moduli found for GSH-Au can be even higher than those for GSH-CdTe NPs of similar size (Fig. 5b, c vs Fig. 1h, i Hydrogel-590). The increase of storage moduli for

GSH-Au is associated with greater strength of the van der Waals interactions between the Au NPs than between CdTe NPs; the characteristic Hamaker constants for particles made from gold and CdTe in water are $13 \times 10^{-20}$ J and $4.9 \times 10^{-20}$ J, respectively. However, the VFOM value is 0.21 MPa at 10 Hz for GSH-Au hydrogels with a size of 3.0 nm of Au NPs, which is considerably lower than 3.2 nm GSH-CdTe hydrogels (Hydrogel-590) which is 1.71 MPa. These finding are significant because they clearly show that both core-to-core and shell-to-shell interactions are significant for the mechanics of these NPs gels with relatively short surface ligands.

CYS-CdTe and MPA-CdTe hydrogels display higher storage and loss moduli ($G''$ and $G''$) when the viscoelastic solids are made from larger particles (Fig. 5d–i). These observations are commensurate with the fact that the molecular geometry of the bonding of CYS and MPA to NPs was found to change little with size. The positions of the NMR signals are commensurate with the two-point binding of both CYS and MPA to CdTe (Supplementary Figure 17)[41,42]. A comparison of stiffness for all three types of surface ligands, that is, GSH, CYS, and MPA, with CdTe NPs of similar size, shows that MPA-CdTe has the highest stiffness (Supplementary Figure 18), which confirms the significance of the supramolecular interactions between the surface ligands for gel mechanics. Moreover, the storage and loss moduli found for 3.2 nm CYS-CdTe hydrogels (Fig. 5d–f vs. Fig. 1g–i) can be even higher than those for GSH-CdTe. These experimental data substantiate the expectations that stiffness and energy dissipation in these systems can be increased in parallel. As expected, the VFOM value is 0.10 MPa at 10 Hz for CYS-CdTe hydrogels with a size of 3.2 nm of CdTe NPs, which is comparable to that of MPA-CdTe hydrogels with a size of 3.1 nm of CdTe NPs with VFOM = 0.16 MPa at 10 Hz.

**Comparative evaluation of viscoelastic properties.** The previous point can be further strengthened by the comparison of mechanical properties of all NP gels obtained in this work with those of other inorganic, polymeric, biological, and nanostructured gels (Supplementary Table 4). The characteristic structural elements of inorganic gels are particle chains organized into porous networks cross-linked by covalent and non-covalent interactions[6,9,13,43] between the constituent hard particles. Similar structure of porous networks can also be found in many hydrogels from proteins[44–46], polymers[22,47,48], peptides[5], and recently from aramid nanofibers[49,50]. The cross-links between these macromolecules are made via hydrophobic interactions, ionic bridges, and hydrogen bonds[16]. Unlike both inorganic and organic networked gels, NP gels described here are compact solids in which each unit retains its mobility, which leads to greater stiffness and energy dissipation per voxel.

The dense packing of NPs in the gels and the dominance of non-covalent intermolecular interactions unifies them with dense gels from proteins or lipids[51]. However, their macroscale mechanics can involve the reconfiguration of the tertiary structure of the protein globule[52], which results in an earlier onset of flow and lower stiffness. vdW interactions between inorganic cores due to the organic–inorganic duality of the NPs

**Fig. 3** Molecular structure of GSH stabilizers at the CdTe NP surface. [1]H NMR spectra of **a** GSH at pH = 10.0, **b** Cd$_x$(GSH)$_y$ complexes, **c** Hydrogel-544, **d** Hydrogel-590, and **e** Hydrogel-618. Amino acid residues in GSH are indicated by a single-letter code: Glutamyl, Q; Glycinyl, G; Cysteinyl, C. **f** Two-dimensional [1]H–[1]H ROESY spectrum of Hydrogel-544. Schematic representations of **g** the three-point bonding (TPB) mode and **h** the single-point bonding (SPB) mode between GSH and the edge and face of a CdTe NP, respectively. The two bonding modes are clearly visible in the NMR spectra (**b-e**) as indicated by the symbol (star) for TPB and symbol (triangle) for SPB. Measurements of the density of GSH indicate that only one layer of GSH is present around each NP, so ligands attached to the particle surfaces are also participating in the gel network

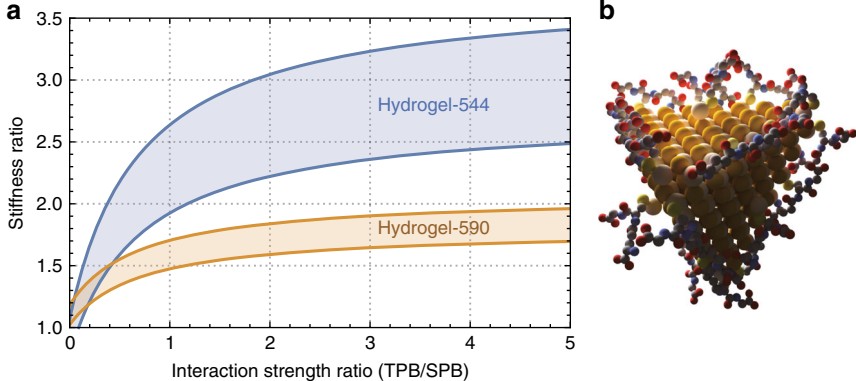

**Fig. 4** Molecular origin of NP hydrogel mechanics. **a** Theoretical bounds for the stiffness of Hydrogel-544 (blue) and Hydrogel-590 (orange) compared to Hydrogel-618 as a function of the interaction strength ratio between TPB–GSH and SPB–GSH. If TPB-mediated interactions between NPs are twice as strong as SPB-mediated ones and mainly present in the edges (upper bound), a threefold increase in stiffness is expected for smaller CdTe hydrogels, **b** schematic representation of a 2.7 nm CdTe NP showing three-point bonds covering corners and edges and single-point bonds are covering the surfaces. Our models suggest that the higher stiffness in hydrogels from smaller NPs results from the relative higher number of three-point bonds. Note: GSH Colors follow legend in Fig. 3

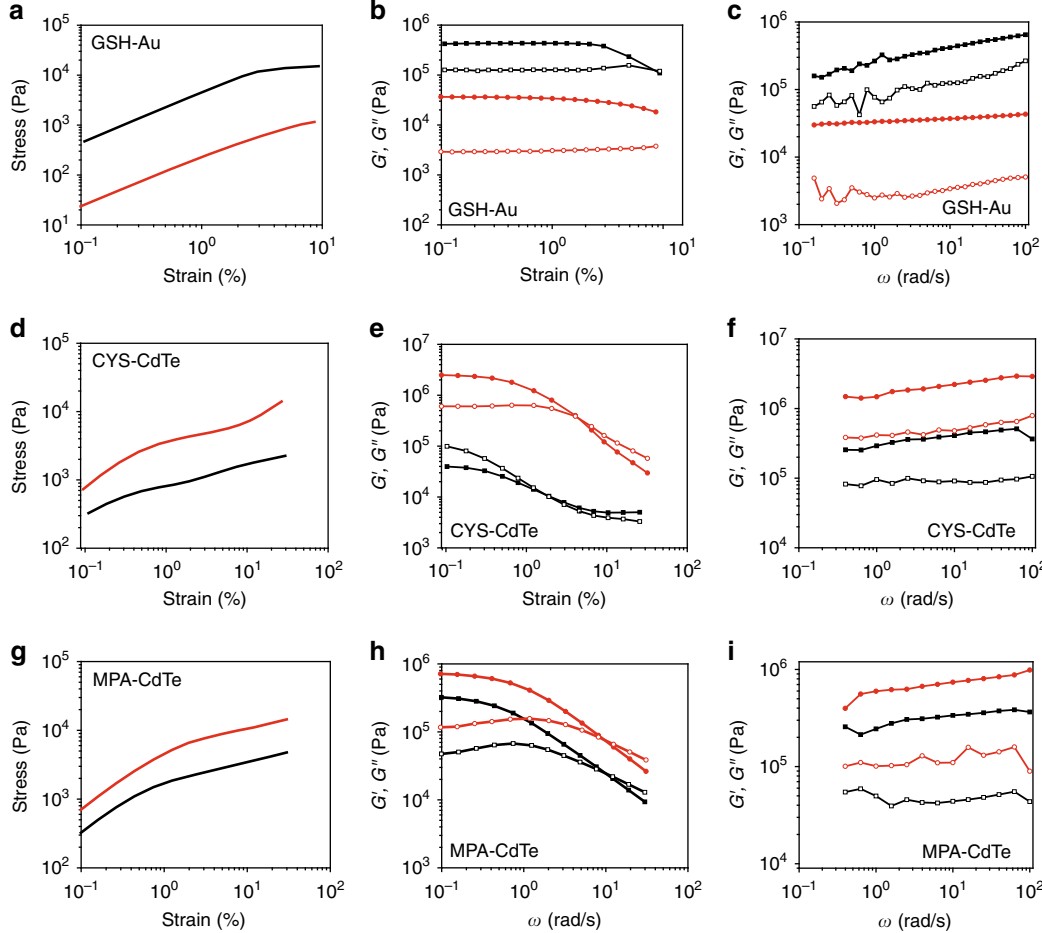

**Fig. 5** Mechanical characterization of GSH-Au hydrogels, CYS-CdTe hydrogels and MPA-CdTe hydrogels. Oscillatory stress/strain (**a**), continuous step moduli/strain (**b**) and rheological dynamic oscillatory frequency sweep (**c**) of GSH-Au hydrogels (black line: Au, ~ 3 nm, red line: Au, ~ 8 nm). Oscillatory stress/strain (**d**), continuous step moduli/strain (**e**) and rheological dynamic oscillatory frequency sweep (**f**) of CYS-CdTe hydrogels (black line: ~ 3.2 nm, red line: ~ 3.7 nm; solid symbol: $G'$, empty symbol: $G''$). Oscillatory stress/strain (**g**), continuous step moduli/strain (**h**) and rheological dynamic oscillatory frequency sweep (**i**) of MPA-CdTe hydrogels (black line: ~ 2.7 nm, red line: ~ 3.1 nm; solid symbol: $G'(\omega)$, empty symbol: $G''(\omega)$)

increase the inter-unit attraction and therefore stiffness (storage modulus) of NP gels. As such, the storage moduli of NP gels is similar to those of covalent lamellar hydrogels from graphene[53] and 10 times higher than the storage moduli of covalent NP gels[9,42].

In summary, this study demonstrates unique gel mechanics of biomimetic NPs with short surface ligands that combine strong supramolecular interactions and strong attractive forces specific to inorganic matter. As examples of their unusual viscoelastic properties, the simultaneous increase of stiffness and energy dissipation of GSH-CdTe, a combination of properties that has proven difficult to achieve for traditional gels and other materials, was demonstrated. VFOM values for GSH-CdTe and other NP gels studies here exceeded those for NP and other gels by several orders of magnitude. A generalizable framework for the interactions between biomimetic NPs combining both shell-to-shell and core-to-core interactions at molecular and nanoscale, respectively, offers itself for further parameterization into realistic computational models[23,26,30], for description of the gel mechanics at macroscale, and in silico design of NP gels.

## Methods

**Chemicals**. L-glutathione (L-GSH) (reduced), Chloroauric Acid, L-cysteine (L-CYS) hydrochloride, and mercaptopropionic acid (MPA) were purchased from Sigma-Aldrich. Cadmium perchlorate hexahydrate and sodium borohydride were obtained from Alfa-Aesar. Aluminum telluride powder was purchased from Materion Advanced Chemicals. All chemicals are used as received.

**Preparation of L-GSH-CdTe NPs**. The preparation of L-GSH-CdTe NPs followed the method of arrested precipitation[54]. Briefly, 0.019 M of $Cd(ClO_4)_2 \cdot 6H_2O$ and 0.025 M of L-GSH were dissolved in 100 ml deionized water. Then the pH of the solution was adjusted to 10.0 by adding 2 M NaOH. The solution was placed into a three-necked flask and bubbled with nitrogen for about 60 min. Subsequently, $H_2Te$ generated by mixture of 0.01 M $Al_2Te_3$ and 10 ml 0.5 M sulfuric acid was carefully injected into the solution. The solution turned from colorless to red. CdTe NPs with different sizes were obtained by heating in a 100 °C oil bath for 1, 3, and 6 h.

**Preparation of GSH-CdTe gels**. Isopropanol and freshly prepared GSH-CdTe solution were mixed in a 2.5:1 volume ratio, and then centrifuged at 5000 r.p.m./min for 5 min. The viscous transparent NPs drop was directly extracted from the bottom of the solution and then dried. Residual isopropanol was removed in a vacuum desiccator for 24 h at room temperature. GSH-CdTe hydrogels were obtained by adding 22 w/w % ultra-pure water into the dried samples. All property tests were conducted after a maturation time of 8 h.

**Preparation of MAP-CdTe, CYS-CdTe, GSH-Au NP gels**. The preparation of MPA- CdTe and CYS-CdTe NPs are followed the procedure of preparation of L-GSH-CdTe NPs. The preparation of GSH capped Au NPs were followed reported procedure[55,56]. The preparation of all hydrogels are followed the method of preparation of GSH-CdTe gels.

**Freeze-drying of hydrogels**. All hydrogels were frozen in liquid nitrogen for 15 min, and subsequently transferred to a cooling condenser at −80 °C (VIRTIS Genesis freeze-dryer; VirTis, SP Industries, Gardiner, NY, USA) until all samples were totally dry.

**UV–vis absorption spectra** were carried out on an 8453 UV–vis Chem Station spectrophotometer produced by Agilent Technologies. Fluorescence spectra were collected on Horiba Fluoro MAX-3. UV–vis absorption spectroscopy provides a preliminary evaluation of intact GSH on the surface of NPs in the gelation process via adding isopropanol. As a comparison of partly removal of GSH process with our gelation method, we separately added 1:1 and 2.5:1 volume ratio of isopropanol to a fresh CdTe solution with photoluminescence (PL) maxima peak at 523 nm (NPs-523), 566 nm (NPs-566), 600 nm (NPs-600), and 647 nm (NPs-647). This induced precipitation of CdTe NPs and CdTe NPs viscous liquid respectively. We re-dispersed the NPs precipitation and viscous liquids with the same volume water in order to make sure both solutions have equal concentration. As shown in Supplementary Figure 1, GSH-CdTe NP solutions were prepared by adding 2.5:1 ratio for isopropanol to solution. The experiment showed stronger UV absorption intensity in the region of 200–300 nm than the re-dispersed solution of GSH-CdTe precipitation, which means the GSH-CdTe hydrogel contains more stabilizers.

The relative quantum yields of the NPs were calculated by comparing it to Rhodamine 6G with a known quantum yield ~ 95 % with 480 nm excitation in ethanol using the equation $Y_{NP} = 0.95 \ F_{NP} \times A_s / F_s \times A_{NP}$, where $F_s$ is the integrated intensity and $A_s$ the optical density.

**Rheological measurements** were performed on a strain controlled Rheometrics RFS II (TA Instruments, New Castle, Delaware). The rheometer is equipped with a parallel fixed plates geometry (diameter 25 mm) and was uses with a 2 mm gap at 25 °C. Hydrogels were poured directly onto the plate of the instrument. The top plate was carefully lowered at a rate of $3.0 \times 10^{-3}$ mm/s for 600 s (10 min) so as not to disrupt the structure of the NP Hydrogel. The normal force on the upper plate during loading never exceeded $6.00 \times 10^2$ grams (~6 Newtons). The samples were allowed to stand until the force practically dissipated to zero. In order to reduce the test noise owing to thermal agitations of NPs, hydrogels were used only after 24 h storage. The samples were protected from drying by a silicon oil cover to prevent the evaporation of water. This protection ensured sample stability over a time period long enough (i.e., 3 h) to perform the measurements of the shear mechanical properties. Frequency dependencies for oscillatory shear deformations of the GSH-CdTe hydrogels were characterized using a fixed strain of 0.01 %. The viscoelasticity and shear dynamics for each NP gel were measured and characterized with an oscillatory strain sweep test carefully performed from low to high-strain starting at 0.01–20 % at a frequency of 1 Hz or 6.28 rad/s with the NP hydrogel loaded between the plates.

The viscoelastic properties of materials are characterized by the viscoelastic figure of merit (VFOM), we calculate VFOM following the method developed by R. Lakes and coworkers in ref. [57]. Namely, we used VFOM = $|G^*| \tan\delta$, $|G^*| = (G'^2 + G''^2)^{0.5}$, where $G^*$ is the complex dynamic modulus, $G'$ is the storage modulus and $G''$ is the loss modulus.

**Calculation of the size of CdTe NPs**. The calculation of the size of CdTe NPs was based on the equation (1) in ref. [58]

$$\Delta E_g = E_{g,QD} - E_{g,0} = a_1 e^{-d/b_1} + a_2 e^{-d/b_2} \qquad (1)$$

where $E_{g,QD}$ is the band gap of NPs obtained from the wavelength of the first excitonic absorption peak, and $E_{g,0}$ is the bulk band gap. The first excitonic absorption peaks for all hydrogels were calculated from the absorption of highly diluted hydrogel solutions in order to avoid the redshift cause by NPs aggregation. In the GSH-NP hydrogels, the first excitonic absorption peak is 472 nm, 520 nm, 546 nm and 587 nm for Hydrogel-544, Hydrogel-590, Hydrogel-618, and Hydrogel-657. In the CYS-NP hydrogels, the first excitonic absorption peak is 520 nm and 550 nm. In the MPA-NP hydrogels, the first excitonic absorption peak is 477 nm and 500 nm. For CdTe NPs, $E_{g,0} = 1.61$ eV, $d$ is the diameter of the CdTe NPs, $a_1 = 5.77$, $b_1 = 8.45$, $a_2 = 1.33$, $b_2 = 43.73$.

**STEM images and EDX analysis** were taken using a JEOL 2010F scanning transmission electron microscopy in annular dark-field mode at 200 kV. SEM images were collected by FEI Nova 200 Nanolab. NMR spectra were collected on a Varian 500 MHz NMR spectrometer.

**Porosity analysis**. The nitrogen adsorption isotherms were measured at 77 K on a Surface Characterization Analyzer (3Flex) from Micromeritics. About 1.3 g of the dried GSH-CdTe hydrogels powder was transferred into the measuring cell and degassed about 24 h at 323 K under vacuum before the adsorption and desorption measurement. The surface area of the sample was calculated by using BET equation ($0.05 < p/p_0 < 0.3$), and the distribution of the pore size was determined from the adsorption or desorption branches of the isotherm using BJH theory. The quantity of adsorbed gas are < 0.01 cm³/g for all three GSH-CdTe hydrogels. Correspondingly, the calculated BET surface area for hydrogels are <0.1060 m²/g. It means the structure of GSH-CdTe hydrogels are solid NP aggregations containing randomly distributed low-density pore from hundreds nanometers to micrometers.

**X-ray photoelectron spectra (XPS)** provides further support of the assumption derived from NMR results (Supplementary Figure 15). To get a more obvious comparison for the ratio of three-point bond (TPB) and single-point bond (SPB) configurations, we investigated the XPS of Hydrogel-544 and Hydrogel-618. Comparing the Cd 3d region of Hydrogel-544 with Hydrogel-618, we observe that the coordination of Cd in Hydrogel-544 is more complex than that in Hydrogel-618. This is in accord with an abundance of $NH_2$ and COO bonding with Cd on the surface of CdTe NPs in Hydrogel-544 compared to Hydrogel-618. Similar results can be found in the N 1s energy level region. The asymmetric spectra in Hydrogel-544 can reasonably well be attribute to N–C covalent and N–Cd coordination. The spectra of Hydrogel-618, however, indicate symmetric N–C covalent. For the oxygen energy level region it is hard to give a detailed attribution due to its complex coordination.

**Weight loss calculation**. Thermogravimetric analysis was performed to evaluate the volume ratio of hydrogel on a Perkin Elmer Instrument Pyris 1 (Supplementary Figure 6). The amples were analyzed in platinum pans at a heating rate of 10 °C per min up to 900 °C in an atmosphere of air flowing at 180 mL/min. To completely burn out L-GSH on the surface, the temperature was kept at 600 °C for 30 min. All three samples have a slightly weight increase at high temperature > 700 °C, possibly due to oxidation of Cadmium in the atmosphere. The approximate weight percent of GSH in GSH-CdTe corresponding to Hydrogel-544, Hydrogel-590, and Hydrogel-618 is ~ 47.3 %, ~ 40.6 %, and ~ 31.7 %, respectively. Correspondingly,

the volumetric ratio of the soft GSH shell vs. the CdTe core in Hydrogels is about 3.7:1 for Hydrogel-544, 2.8:1 for Hydrogel-590, and 1.9:1 for Hydrogel-618.

**Methodology for theoretical modeling**. *Covering density of NP surfaces with GSH:* We calculate the average absolute number of GSH molecules per CdTe NP, $n_{GSH}$, from the equation (2)

$$\text{Weight ratio} = \frac{\text{GSH weight \%}}{\text{CdTe weight \%}} = n_{GSH}\frac{w_{1GSH}}{w_{1CdTe}} \quad (2)$$

where GSH weight % = (47.3%; 40.6%; 31.7%) for Hydrogel-544, Hydrogel-590, and Hydrogel-618, respectively, CdTe weight % = 1−GSH weight %, $w_{1GSH} = 5.1 \cdot 10^{-22}$ g, and $w_{1CdTe} = (135; 226; 349) \cdot 10^{-22}$ g. We obtain:

$$\text{Weight ratio} = (0.897; 0.683; 0.464), \quad (3)$$

$$n_{GSH} = (23.7; 30.3; 31.7). \quad (4)$$

For brevity here and below, we shall use the notations (X, Y, Z) to refer to the set of parameters characteristic of Hydrogel-544, Hydrogel-590, and Hydrogel-610, respectively. Normalized per surface area, the density of GSH is (13.0; 11.8; 9.3) molecules per nm². Using the ratio between SPB and TPB configuration obtained from NMR measurements, $n_{SPB}/n_{TPB}$=(0.2; 0.4; 0.7), we get

$$n_{TPB} = (19.8; 21.9; 18.6) \text{ and } n_{SPB} = (3.9; 8.8; 13.1), \quad (5)$$

or (10.9; 8.4; 5.4) GSH-TPB per nm² and (2.2; 3.4; 3.8) GSH-SPB per nm² for Hydrogel-544, Hydrogel-590, and Hydrogel-618, respectively.

*Lower and upper stiffness bounds:* Decreasing the NP size by a factor of l causes the total available NP surface area per volume of hydrogel to increase by the same factor l and the available edge length to increase by a factor $l^2$. Here we assume that the NP packing density remains constant. If the density of TPB–GSH and SPB–GSH were constant across all NP hydrogels, we would therefore expect the stiffness k of the hydrogel to grow between linearly and quadratically with decreasing NP size. Although the determination of an exact value based on atomistic calculations is beyond the scope of this work, we can create these bounds explicitly using available experimental data.

Not only is the density of GSH ligands different for different hydrogels, but also there is no reason to believe that SPB–GSH and TPB–GSH contribute equally to NP–NP binding. We expect that the stiffer, more compact geometry of TPB–GSH molecules produces more exposed hydrogen bonds, resulting in a stronger binding between neighboring particles. We arrive at the following formula (6):

$$\left(\frac{k_1}{k_2}\right) = \left(\frac{l_1}{l_2}\right)^i \left(\frac{n_{TPB,1}}{A_1}\varepsilon + \frac{n_{SPB,1}}{A_1}\right)\bigg/\left(\frac{n_{TPB,2}}{A_2}\varepsilon + \frac{n_{SPB,2}}{A_2}\right) = \left(\frac{l_2}{l_1}\right)^i \frac{n_{TPB,1}\varepsilon + n_{SPB,1}}{n_{TPB,2}\varepsilon + n_{SPB,2}}. \quad (6)$$

Here i is a dimensionality factor referring to the NP size scaling (i = 1 for surface-mediated interactions, and i = 2 for edge-mediated interactions), l is the NP edge length, A the NP surface area, $\varepsilon$ the relative bond strength of TPB compared to SPB, and the index 1, 2 corresponds to two distinct NP sizes. The first term corresponds to the difference in available surface area/edge length between NPs. The second and third terms represent the ratio between the ligand density scaled by the difference in strength between TPB–GSH and SPB–GSH. We plot in Fig. 4a the two bounds for hydrogel-544 (blue) and hydrogel-590 (orange) relative to hydrogel-618.

**Visualizations of GSH on a CdTe NP** were constructed from energy minimized configurations obtained from the molecular software Spartan Wavefunction[59]. All NPs are in scale with the appropriate number of atoms corresponding to the NP sizes measured in experiments. The TPB–GSH model was determined by a simple SEMI-EMPIRICAL simulation using the software package Spartan (Wavefunction Inc., Irvine, CA). In the simulations two of different binding model of GSH on the edge of the CdTe NP were considered. Model I includes three of Cd atoms which is S (Cys)-Cd, N (Glu)-Cd, and COO (Glu)-Cd, Model II involves S (Cys)-Cd and N (Glu)-Cd-OOC (Glu). For simplicity, we used a stoichiometric composition of $C_{10}H_{49}N_3O_6Cd_{13}Te_{11}$ as the atomic model and assumed that all the unsaturated Cd and Te ions are capped by the –H groups. The optimization algorithm varied the bond lengths and angles to find the lowest energy structures of GSH-CdTe NP. The energies represent the sum of electronic, vibrational, rotational, nuclear, and translational energy components for a specific atomic model. The energy of Model I in the defined equilibrium geometry is 572.0 kJ/mol, while in Model II is 682.3 kJ/mol. Therefore, we hypothesized that GSH tended to bind with CdTe with three of Cd atoms in the TPB model (Supplementary Figure 14).

The three-dimensional rendering shown in Fig. 4b (main text) was obtained using the open source computer graphics software Blender[60].

**Data availability**. All relevant data are available from the authors upon request.

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

## Acknowledgements

This work was supported by the Army Research Office (ARO) under MURI project W911NF-10-1-0518, Reconfigurable Matter from Programmable Colloids, and the EFRI-BSBA 0938019 project from the National Science Foundation (NSF). The MURI grant supported in particular the collaboration between the Kotov and Glotzer groups in the development of the theoretical model. Y.Z. acknowledges support from the National Natural Science Foundation of China (21573162, 21773172) and WIBEZD2014001-02. Partial support for N.A.K. and S.C.G. was also provided by the Center for Photonic and Multiscale Nanomaterials (C-PHOM) funded by the NSF Materials Research Science and Engineering Center program DMR 1120923. N.A.K., Y.Z., and J.Z. were supported by NSF projects CBET 1036672, CBET 1403777; and DMR 1411014. The authors thank Prof. Xueqian Kong and Dr Jun Zhang from the Zhejiang University for assisting the NMR experiments and fruitful NMR discussions.

## Author contributions

Y.Z. conducted the experimental set-up and measurements. P.F.D., M.E. and S.C.G. established the simulation model. B.S.S., R.H. and A.R. carried out the NMR study. K.J., C.M. and P.F.G. contributed to the viscoelasticity analysis. K.S. helped with STEM. J.Z., M.Y. and F.T. discussed STEM images and viscoelasticity data. N.A.K. conceptualized the project. Y.L.Z., P.F.D., M.E., B.S.S. and N.A.K. wrote the manuscript.

## Additional information

**Competing interests:** The authors declare no competing financial interests.

