## [Peer Review File · Nature Communications]

Reviewers' comments:

Reviewer #1 (Remarks to the Author):

As requested, I have evaluated the revised manuscript "Unusual Molecular and Macroscale Mechanics of Nanoparticle Hydrogels", originally submitted to [Redacted] and now being considered for Nature Comm, as well as the supplementary information, and the authors' response to this reviewers original critiques. My overall sense is that there is some interesting phenomenon here, but the paper fails to make a clear, strong case for nature of the phenomenon and its universality. These concerns may be addressed with additional data, as noted below.

The key hypothesis is that when the ratio of triple-point bonding (tpb) to single-point bonding (spb) is increased, the storage and loss moduli increase, resulting in an increase in the figure of merit (VFOM). Moreover, the presence of tpb leads to unusually large moduli. The authors provide clear data showing that changing the nanoparticle size changes the ratio of tpb to spb, with the smallest particles exhibiting the largest ratio, and this is reflected in increased storage and loss moduli and increased VFOM. Hence the unexpected observation of smaller colloidal particles in the gel (and therefore a higher ratio of soft ligand component to hard colloid component) leading to higher strength. This is attributed to the ability of the tpb ligands to lie flat on the surface of the particles, facilitating stronger interparticle interactions. The same size trend in moduli is observed with the Au particles, but the VFOM is considerably smaller, attributed to the greater van der Waals interactions between Au relative to CdTe.

To fully support the hypothesis that it is the ratio of the tpb/spb that governs the moduli in GSH-modified colloidal gels, the authors need to provide these ratios for the Au particles so as to confirm that smaller particles do, in fact, have higher ratios. This data can be acquired from proton NMR. They should also provide VFOM data for the two sizes, as the size trend there should also be similar (only data for the 3 nm particles is presented).

With respect to concerns about possible morphological effects, the authors have performed SAXS and BET surface area data. The former is shown in the response to reviewers and the latter is stated to be very similar. These data should be provided in the supplementary information and referred to in the main text.

In Figure 3 g and h and Figure 4a, please provide a legend for the atom colors in the molecule. Which are C, N, O, S, H? it is not clear. For Figure 3, the top surface should be expanded so that the molecule is clear. Most of the figure comprises the repeating structure of CdTe, which is hardly relevant. For Figure 4, the model implies that the particles are pyramidal in shape, but no evidence is provided that this is actually the case. If the authors really want to imply that edges, etc. are where the tpbs are located (and not just highly curved surfaces of small particles) they need to be looking at different morphologies. The discussion (and associated images) should be constrained to what they actually know (i.e., the ratio of tpb to spb). If there is speculation, it would be more usefully applied to understanding how the molecules on the surface hydrogen bond to each other to contribute to the gel formation. The statement that it is identical to how they bind to the nanoparticle surface implies that a single molecule is bridging two particles. If this is the point the authors are trying to make, they should be more clear about it. If instead intermolecular forces are contributing to gel formation, please show us how this might be occurring.

Right now, the impact of the paper is limited by the fact that the tpb is a phenomenon that has only been demonstrated with a single ligand, GSH. If later work shows this to be general (i.e., ligands can be designed or chosen that promote this bonding), the paper is likely to be impactful. If it turns out that GSH is just a one-off, the impact will depend on the value of the hydrogels with their unique and unexpected properties.

Responses to Reviewers' Comments.

We would like to thank Reviewer's time and effort in evaluating our manuscript. Please find below the point-by-point replies to the insightful remarks. We hope that the changes made in the manuscript address them in detail.

Reviewer #1

Comment 1-1: To fully support the hypothesis that it is the ratio of the tpb/spb that governs the moduli in GSH-modified colloidal gels, the authors need to provide these ratios for the Au particles so as to confirm that smaller particles do, in fact, have higher ratios. This data can be acquired from proton NMR. They should also provide VFOM data for the two sizes, as the size trend there should also be similar (only data for the 3 nm particles is presented).

Response 1-1: We thank for reviewer's comments and suggestions. We acquired the requested data. As predicted, the ^1H NMR of GSH-Au indicated that GSH bond to Au with two configurations (Figure.S14). It also showed configuration transformation with increasing the size of Au from 3 nm to 8 nm. We assumed that the configuration transformation is TPB to SPB based on ^1H NMR and corresponding ^1H - ^1H COSY NMR. The ratio of TPB/SPB for 3 nm of GSH-Au hydrogel and 8 nm of GSH-Au hydrogel is 1:0.1 and 0.26:1 respectively.

The VFOM data for GSH-Au, MPA-CdTe and CYS-CdTe with two sizes were also listed in Table S3. It should be noted that the VFOM value for 8 nm of GSH-Au is quietly lower than for other nanoparticle hydrogels due to their much bigger size.

Comment 1-2: With respect to concerns about possible morphological effects, the authors have performed SAXS and BET surface area data. The former is shown in the response to reviewers and the latter is stated to be very similar. These data should be provided in the supplementary information and referred to in the main text.

Response 1-2: We thank the reviewer for the suggestion. We have added the SAXS and BET analysis to supplementary information and corresponding discussions in the main text.

Comment 1-3: In Figure 3 g and h and Figure 4a, please provide a legend for the atom colors in the molecule. Which are C, N, O, S, H? it is not clear. For Figure 3, the top surface should be expanded so that the molecule is clear. Most of the figure comprises the repeating structure of CdTe, which is hardly relevant.

Response 1-3: We appreciate the comment and the requested changes were made.

Comment 1-4: For Figure 4, the model implies that the particles are pyramidal in shape, but no evidence is provided that this is actually the case. If the authors really want to imply that edges, etc. are where the tpbs are located (and not just highly curved surfaces of small

Figure R1. a) Tetrahedral CdTe nanoparticle, Yeom *et al. Nature Materials* 2015,14, 66-72. b) Tetrahedral CdTe nanoparticle Tang *et al. Science* 2006, 314,274-278. c) Tetrahedral CdSe nanocrystals. Yang *et al. Angew. Chem.* 2005, 117, 6870 –6873. d). Tetrahedral CdSe nanocrystals. Boles M. A. and Talapin D. V.J. *Am. Chem. Soc.* 2014, 136, 5868–5871.

particles) they need to be looking at different morphologies. The discussion (and associated images) should be constrained to what they actually know (i.e., the ratio of tpb to spb). If there is

speculation, it would be more usefully applied to understanding how the molecules on the surface hydrogen bond to each other to contribute to the gel formation. The statement that it

is identical to how they bind to the nanoparticle surface implies that a single molecule is bridging two particles. If this is the point the authors are trying to make, they should be more clear about it. If instead intermolecular forces are contributing to gel formation, please show us how this might be occurring.

Response 1-4: We agree with the referee that the structural data could always be more detailed. The addition of the NMR data will add more clarity to the matter. The wide spectrum of the NPs used in the study also confirm the conclusion made about the molecular reasoning behind the mechanical properties of the NP gels.

With respect to the experimental evidence of the tetrahedral shape, several works prior to this one have provided evidence that small nanoparticles with zinc blende structure have tetrahedral shape (**Figure R1**, previous page). Therefore the assumption of the tetrahedral shape is factually justified. To address the comment of the Reviewer, we have added these references to the manuscript.

The justification for the assumption that triple-point bonding (TPB) model provides a stronger gel than single-point bonding (SPB) stems directly from experiments (section 3.1 of SI). It is also supported by the (simple) model to estimate a relationship between the strength of the gels and the degree by which TPB ligands are attached stronger than SPB ligands (section 3.2 in the SI).

We separate the experimental facts from speculations threatening the geometry of the surface ligands. Stiffness of the gels, plotted in **Figure 4a**, can be calculated by assuming that the ligand-mediated interactions are mostly edge-mediated (**Figure 4a**, upper bounds) or facet-mediated (**Figure 4a**, lower bounds). Both lines are shown and our plot only depends on the assumption that TPB ligands cause a stronger interparticle connection than SPB ligands, whatever the nature of such interconnectivity may be.

With respect to the location of the ligands, this is relevant when trying to answer a second puzzle that emerges from the experimental data, namely *why is the number of TPB ligands nearly constant when increasing the size of CdTe NPs whereas the number of SPB ligands vastly increases*. One natural possibility is that TPB ligands attach primarily to vertices and edges of nanoparticles, which grow more slowly than the facets, as NP size is increased. This possibility would push us towards the upper bounds in **Figure 4a**, suggesting that small changes in particle sizes for CdTe nanoparticles can have a dramatic effect in the stiffness of hydrogels, as measured by our experiments.

We rewrote the section related to these points and hope the new explanation is clearer and the assumptions made more explicit. New quantum mechanical data regarding the molecular configuration of TPBs were also added in Supplementary Information, Section 3.

Comment 1-5: Right now, the impact of the paper is limited by the fact that the tpb is a phenomenon that has only been demonstrated with a single ligand, GSH. If later work shows this to be general (i.e., ligands can be designed or chosen that promote this bonding), the paper is likely to be impactful. If it turns out that GSH is just a one-off, the impact will depend on the value of the hydrogels with their unique and unexpected properties.

Response 1-5: We sincerely appreciate and agree reviewer's suggestions. In this work we have investigated nanoparticle gels including three ligands demonstrating ligand re-configurations. This indicates the thoroughness of the approach. Several NP cores demonstrate generality of the phenomenon. We agree that there are still plenty of work in this field deserving to investigate. However, the introduction of another peptide WILL require the new NMR study, other spectroscopic studies, gel mechanics experiment, and the new model (see Comment 1-1 and Reply1-1). We strongly agree that it will be useful, but it is better placed in a separate study.

In our current work, we are systematically investigating peptides capped semiconductor nanoparticles and metal nanoparticles, we are trying to control the interface structure between peptides and nanoparticle core as well as increasing the VFOM value of peptide-nanoparticle gels. We believe that the cooperative interaction forces between ligands and inorganic core will enable the ligands adjusted nanoparticle hydrogels to show both expected and unexpected mechanical properties.

REVIEWERS' COMMENTS:

Reviewer #1 (Remarks to the Author):

The authors have addressed the issues brought up in previous review cycles. Acceptance is recommended.